# Development and validation of a risk prediction model for severe postoperative complications in elderly patients with hip fracture

Zhihui Wei[1‡], Lian Jiang[2‡], Minghua Zhang[1], Xiao Chen◯[3]*

1 Department of Orthopedics, Yongchuan Hospital of Chongqing Medical University, Yongchuan, Chongqing, China, 2 Department of Geriatrics, Yongchuan Hospital of Chongqing Medical University, Yongchuan, Chongqing, China, 3 Department of Orthopedics, The First People's Hospital of Neijiang, Neijiang, Sichuan, China

‡ ZW and LJ are co-first authors.
* 342918992@qq.com

**Data Availability Statement:** All relevant data are within the manuscript and its Supporting Information files.

## Abstract

### Objective

This study aimed to investigate risk factors associated with severe postoperative complications following hip fracture surgery in elderly patients and to develop a nomogram-based risk prediction model for these complications.

### Methods

A total of 627 elderly patients with hip fractures treated at Yongchuan Hospital of Chongqing Medical University from January 2015 to April 2024 were collected. 439 patients were assigned to the training cohort for model development, and 188 to the validation cohort for model assessment. The training cohort was stratified based on the presence or absence of severe complications. We employed LASSO regression, as well as univariate and multivariate logistic regression analyses, to identify significant factors. A nomogram was constructed based on the outcomes of the multivariate regression. The model's discriminative ability was assessed using the area under the receiver operating characteristic curve (AUC), while calibration plots and decision curve analysis (DCA) evaluated its calibration and stability. Internal validation was performed using the validation cohort.

### Results

Out of the 627 patients, 118 (18.82%) experienced severe postoperative complications. Both LASSO regression and multivariate logistic analysis identified the modified 5-item frailty index (mFI-5) and the preoperative C-reactive protein to albumin ratio (CAR) as significant predictors of severe complications. The nomogram model, derived from the multivariate analysis, exhibited strong discriminative ability, with an AUC of 0.963 (95% CI: 0.946–0.980) for the training cohort and 0.963 (95% CI: 0.938–0.988) for the validation cohort.

**Funding:** The author(s) received no specific funding for this work.

**Competing interests:** The authors have declared that no competing interests exist.

Calibration plots demonstrated excellent agreement between the nomogram's predictions and actual outcomes. Decision curve analysis (DCA) indicated that the model provided clinical utility across all patient scenarios. These findings were consistent in the validation cohort.

## Conclusions

Both the mFI-5 and CAR are predictive factors for severe postoperative complications in elderly patients undergoing hip fracture surgery.

## Introduction

Aging in China has led to a significant annual increase in the incidence of hip fractures among the elderly, impacting their health [1]. The elderly population experiencing hip fractures is growing by 30% per decade [2], characterized by severe osteoporosis, reduced physical capabilities, and slower reactions, making them vulnerable to falls. Common types include femoral neck, intertrochanteric, and subtrochanteric fractures, with the former two comprising 40%-50% of cases. The one-year mortality rate for elderly hip fractures ranges from 20% to 30%, and more than half of the patients suffer some form of disability [3,4]. Treatment options include conservative and surgical approaches, with surgery increasingly favored due to lower mortality risk compared to conservative methods [5], which are limited to patients unable or unwilling to undergo surgery. Advances in surgical techniques and materials, such as hip replacement and proximal femoral nail anti-rotation (PFNA), have become prevalent [6–9]. However, perioperative complications stemming from patient frailty, surgical trauma, bleeding, anesthesia, and pain significantly impact recovery. Thus, effective prevention and management of these complications are crucial in treating elderly hip fracture patients.

The majority of elderly hip fracture patients present with more than five chronic internal diseases prior to injury [10], complicating clinical management. Current clinical strategies emphasize personalized care and enhanced recovery after surgery (ERAS) principles, crucial for minimizing complications and promoting early recovery. Given the diversity in underlying conditions and physiological states among patients, individualized assessment to prevent complications is essential. Nomograms serve as valuable predictive tools by integrating various patient risk factors to anticipate complication risks. While previous studies have explored individual risk factors for postoperative complications in elderly hip fracture patients, a comprehensive risk prediction model has not yet been established. This study aims to develop and validate a predictive model and nomogram, offering a framework for assessing complication risks clinically and guiding treatment strategies.

## Materials and methods

### Patient data

The study included elderly patients with hip fractures admitted to the Joint Surgery Department of our institution from January 2015 to April 2024. We accessed the data for research purposes on June 03, 2024. The study was approved by the Ethics Committee of Yongchuan Hospital of Chongqing Medical University (approval number: 2024-kelunshen-61), with informed consent waived due to its retrospective nature.

### Eligibility criteria

Inclusion criteria were as follows: (1) Age 65 years or older; (2) Diagnosis of a single acute hip fracture (including femoral neck and intertrochanteric fractures) due to low-energy trauma; (3) Underwent surgical intervention: Total hip arthroplasty or proximal femoral nail antirotation (PFNA) internal fixation.

Exclusion criteria consisted of: (1) Pathologic fractures; (2) Concurrent fractures at multiple sites; (3) Open fractures; (4) History of old fractures or prior hip surgery; (5) Incomplete clinical records.

Postoperative severe complications included: deep surgical site infections, prolonged mechanical ventilation (>48 hours), unplanned reintubation, acute renal failure, sepsis or septic shock, venous thromboembolism (deep vein thrombosis and pulmonary embolism), cerebrovascular accident, stress ulcer bleeding, myocardial infarction, heart failure, respiratory failure, cardiac arrest, and mortality.

### Clinical data collection

Data were extracted from patient hospital records, encompassing gender, age, body mass index (BMI), fracture location, modified Five-Item Frailty Index (mFI-5) score, intraoperative transfusion status, preoperative levels of C-reactive protein (CRP), C-reactive protein to albumin ratio (CAR), serum albumin, hemoglobin, fibrinogen, uric acid, creatinine, triglycerides, preoperative ultrasound findings, operation duration, American Society of Anesthesiologists (ASA) physical status classification, preoperative fasting duration, interval from injury to surgery, surgical approach, and anesthesia type. The primary outcome was the incidence of severe postoperative complications, with data collection concluding at the date of hospital discharge.

### Statistical analysis

Data from the Affiliated Yongchuan Hospital of Chongqing Medical University were randomly divided into training and validation cohorts at a 7:3 ratio, with comparisons made across various variables. Non-normally distributed data were presented as medians with interquartile ranges. Univariate analysis employed chi-square or Fisher's exact tests for categorical variables and Student's t-test or rank-sum tests for continuous variables. LASSO logistic regression was performed on the training set to identify independent predictors and construct a predictive model for severe complications. Nomogram performance was evaluated using ROC curves and calibration plots, with AUC values ranging from 0.5 to 1 indicating discriminative power. Decision curve analysis (DCA) determined the threshold for predictive net benefit, with statistical significance set at $p < 0.05$. Statistical analyses were conducted using SPSS (version 22.0) and R software (version 4.2.2).

## Results

### Patient demographics and baseline characteristics

This study included 627 elderly patients with hip fractures, among whom 118 (18.82%) experienced severe postoperative complications. Using random number table, 70% of patients were allocated to the training set (439 cases), and 30% to the validation set (188 cases). Statistically significant differences ($P < 0.05$) were observed in surgery duration and the American Society of Anesthesiologists (ASA) physical status classification. However, parameters such as gender, age, Body Mass Index (BMI), fracture location, Modified Frailty Index-5 (mFI-5), intraoperative blood transfusion, preoperative C-reactive protein to albumin ratio (CAR), preoperative hemoglobin (HGB), preoperative fibrinogen, preoperative uric acid, preoperative creatinine,

preoperative triglycerides, preoperative ultrasound findings, preoperative fasting duration, time from injury to surgery, surgical approach, and anesthesia type showed no significant differences (P > 0.05), as detailed in Table 1.

## Construction of the nomogram model

In the training cohort, four indicators showed significant differences: mFI-5 (P < 0.001), HGB (P < 0.05), CAR (P < 0.05), and type of surgery (P < 0.05), as shown in Table 2. Through LASSO regression analysis, these were narrowed down to two potential predictive factors. The coefficients are presented in Table 3, and the regression curve is depicted in Fig 1A. The cross-validation error plot for the LASSO model is shown in Fig 1B, indicating optimal regularization for the most parsimonious model. Fig 2 displays the AUCs for these variables, with mFI-5 and CAR achieving AUCs of 0.594 and 0.944, respectively. Multivariate logistic regression analysis of the training cohort is detailed in Table 4. The final model, integrating two independent predictors (mFI-5 and CAR), is illustrated as a nomogram in Fig 3. Model performance is further illustrated in Fig 4, demonstrating AUCs of 0.963 (95% CI: 0.946–0.980) for the training set and 0.963 (95% CI: 0.938–0.988) for the internal validation set, both surpassing individual predictor AUCs and indicating robust predictive accuracy. Calibration plots in Fig 5A and 5B reveal strong alignment between observed and predicted severe postoperative complications, affirming the nomogram's validity. The study confirms the nomogram's reliability in the validation set, with calibration curves closely approximating the ideal, indicating consistency between predictions and actual outcomes. DCA curves related to the nomogram, as depicted in Fig 6A and 6B, highlight the clinical net benefit across various risk threshold probabilities. The findings underscore the substantial clinical utility of the nomogram, as evidenced by its DCA curves.

## Discussion

To mitigate the incidence of complications following hip fracture surgery in the elderly, surgical intervention has become the predominant treatment approach. THA and PFNA are widely adopted due to their minimally invasive nature and quick recovery times. However, challenges such as declining physical function, poor compliance, prolonged immobilization, and slow recovery among elderly patients have contributed to an increased risk of postoperative complications. These complications primarily encompass infections, lower limb deep vein thrombosis, heart and renal failure, gastrointestinal dysfunction, delirium, and mortality, significantly impacting surgical outcomes [11,12]. Previous studies have indicated that up to 20%-40% of elderly hip fracture patients experience postoperative complications, consistent with the findings of this study [13]. There exist diverse risk factors contributing to postoperative complications. This study specifically identified the mFI-5 and CAR as independent predictors of complications following hip fracture surgery. A predictive model was developed and a nomogram constructed, both of which demonstrated strong discriminatory ability, consistency, and effectiveness upon internal validation. These findings offer valuable insights for clinical practice, facilitating timely interventions to mitigate complication risks and promote expedited postoperative recovery.

There are numerous methods and tools for assessing patient frailty, included Fried Frailty Phenotype (FFP)[14], Clinical Frailty Scale (CFS) [15], FRAIL Index [16], Short Physical Performance Battery(SPPB) [17], Edmonton Frailty Scale(EFS) [18], Modified Frailty Index (mFI), each method has its own emphasis, such as difficulties in implementation, incomplete evaluation, too many evaluation indicators, need long time. mFI was often used to assessment surgical preoperative frailty, it include mFI-11 and mFI-5, mFI-5 is improved from mFI-11,

**Table 1. Patient demographics and baseline characteristics.**

| Characteristic | Cohort | | p-value |
| --- | --- | --- | --- |
| | Training Cohort, N = 439 | Internal Test Cohort, N = 188 | |
| **Gender** | | | 0.872 |
| Female | 296 (67.4%) | 128 (68.1%) | |
| Male | 143 (32.6%) | 60 (31.9%) | |
| **Age** | | | 0.682 |
| Mean ± SD | 78 ± 9 | 78 ± 9 | |
| **BMI** | | | 0.529 |
| Mean ± SD | 20.86 ± 1.36 | 20.93 ± 1.35 | |
| **Classification of fracture** | | | 0.594 |
| Fracture of femoral neck | 276 (62.9%) | 112 (59.6%) | |
| Intertrochanteric fracture | 163 (37.1%) | 76 (40.4%) | |
| **mFI-5** | | | 0.463 |
| Mean ± SD | 0.40 ± 0.69 | 0.44 ± 0.71 | |
| **Blood transfusion** | | | 0.899 |
| No | 419 (95.4%) | 179 (95.2%) | |
| Yes | 20 (4.6%) | 9 (4.8%) | |
| **HGB** | | | 0.379 |
| Mean ± SD | 115 ± 19 | 113 ± 18 | |
| **Fibrinogen** | | | 0.052 |
| Mean ± SD | 4.21 ± 6.57 | 3.57 ± 1.43 | |
| **CAR** | | | 0.699 |
| Mean ± SD | 0.70 ± 0.87 | 0.73 ± 0.93 | |
| **Urea nitrogen** | | | 0.617 |
| Mean ± SD | 6.99 ± 3.40 | 7.15 ± 3.76 | |
| **Creatinine** | | | 0.570 |
| Mean ± SD | 79 ± 85 | 76 ± 65 | |
| **Triglyceride** | | | 0.994 |
| Mean ± SD | 1.27 ± 0.62 | 1.27 ± 0.48 | |
| **Color Doppler Ultrasound** | | | 0.778 |
| Abnormal | 290 (66.1%) | 122 (64.9%) | |
| Normal | 149 (33.9%) | 66 (35.1%) | |
| **Operation time** | | | 0.023 |
| Mean ± SD | 85 ± 23 | 81 ± 21 | |
| **ASA** | | | 0.798 |
| I | 6 (1.4%) | 1 (0.5%) | |
| II | 83 (18.9%) | 35 (18.6%) | |
| III | 315 (71.8%) | 134 (71.3%) | |
| IV | 35 (8.0%) | 18 (9.6%) | |
| **Preoperative fasting time** | | | 0.179 |
| Mean ± SD | 11.06 ± 4.97 | 11.49 ± 2.88 | |
| **Injury-surgery interval** | | | 0.415 |
| ≤24h | 13 (3.0%) | 3 (1.6%) | |
| >24h | 426 (97.0%) | 185 (98.4%) | |
| **Surgical method** | | | 0.257 |
| Internal fixation | 388 (88.4%) | 160 (85.1%) | |
| THA | 51 (11.6%) | 28 (14.9%) | |
| **Anesthesia method** | | | 0.242 |
| General anesthesia | 67 (15.3%) | 22 (11.7%) | |
| Combined spinal and epidural anesthesia | 372 (84.7%) | 166 (88.3%) | |

**Table 2. Comparison of variables between complication group and non-complication group.**

| Characteristics | Training Cohort | | | Internal Test Cohort | | |
|---|---|---|---|---|---|---|
| | Non-complication N = 362 | Complication N = 77 | p-value | Non-complication N = 147 | Complication N = 41 | p-value |
| **Gender** | | | 0.608 | | | 0.729 |
| Female | 246 (68%) | 50 (65%) | | 101 (69%) | 27 (66%) | |
| Male | 116 (32%) | 27 (35%) | | 46 (31%) | 14 (34%) | |
| **Age** | | | 0.074 | | | 0.868 |
| Mean ± SD | 78 ± 9 | 79 ± 8 | | 78 ± 9 | 78 ± 8 | |
| **BMI** | | | 0.406 | | | 0.339 |
| Mean ± SD | 20.88 ± 1.37 | 20.74 ± 1.32 | | 20.98 ± 1.38 | 20.77 ± 1.22 | |
| **Classification of fracture** | | | 0.252 | | | 0.071 |
| Fracture of femoral neck | 234 (65%) | 42 (55%) | | 93 (63%) | 19 (46%) | |
| Intertrochanteric fracture | 128 (35%) | 35 (45%) | | 54 (37%) | 22 (54%) | |
| **mFI-5** | | | <0.001 | | | <0.001 |
| Mean ± SD | 0.29 ± 0.56 | 0.91 ± 0.98 | | 0.27 ± 0.54 | 1.07 ± 0.88 | |
| **Blood transfusion** | | | 0.764 | | | 0.412 |
| No | 346 (96%) | 73 (95%) | | 141 (96%) | 38 (93%) | |
| Yes | 16 (4%) | 4 (5%) | | 6 (4%) | 3 (7%) | |
| **HGB** | | | 0.026 | | | 0.144 |
| Mean ± SD | 116 ± 19 | 111 ± 18 | | 115 ± 17 | 109 ± 21 | |
| **Fibrinogen** | | | 0.980 | | | <0.001 |
| Mean ± SD | 4.21 ± 7.21 | 4.20 ± 1.21 | | 3.36 ± 1.38 | 4.30 ± 1.40 | |
| **CAR** | | | <0.001 | | | <0.001 |
| Mean ± SD | 0.46 ± 0.68 | 1.81 ± 0.79 | | 0.47 ± 0.85 | 1.65 ± 0.57 | |
| **Urea nitrogen** | | | 0.302 | | | 0.907 |
| Mean ± SD | 6.89 ± 3.20 | 7.42 ± 4.21 | | 7.13 ± 3.81 | 7.21 ± 3.62 | |
| **Creatinine** | | | 0.338 | | | 0.596 |
| Mean ± SD | 78 ± 87 | 87 ± 70 | | 75 ± 71 | 79 ± 38 | |
| **Triglyceride** | | | 0.744 | | | 0.421 |
| Mean ± SD | 1.27 ± 0.65 | 1.25 ± 0.45 | | 1.25 ± 0.48 | 1.32 ± 0.49 | |
| **Color Doppler Ultrasound** | | | 0.069 | | | 0.104 |
| Abnormal | 246 (68%) | 44 (57%) | | 91 (62%) | 31 (76%) | |
| Normal | 116 (32%) | 33 (43%) | | 56 (38%) | 10 (24%) | |
| **Operation time** | | | 0.771 | | | 0.300 |
| Mean ± SD | 85 ± 23 | 84 ± 23 | | 80 ± 21 | 84 ± 24 | |
| **ASA** | | | 0.135 | | | 0.595 |
| I | 5 (1%) | 1 (1%) | | 1 (1%) | 0 (0%) | |
| II | 75 (21%) | 8 (10%) | | 30 (20%) | 5 (12%) | |
| III | 255 (70%) | 60 (78%) | | 102 (69%) | 32 (78%) | |
| IV | 27 (7%) | 8 (10%) | | 14 (10%) | 4 (10%) | |
| **Preoperative fasting time** | | | 0.349 | | | 0.288 |
| Mean ± SD | 10.86 ± 2.32 | 12.02 ± 10.76 | | 11.37 ± 2.86 | 11.92 ± 2.96 | |
| **Injury-surgery interval** | | | >0.999 | | | >0.999 |
| ≤24h | 11 (3%) | 2 (3%) | | 3 (2%) | 0 (0%) | |
| >24h | 351 (97%) | 75 (97%) | | 144 (98%) | 41 (100%) | |
| **Surgical method** | | | <0.001 | | | 0.053 |
| Internal fixation | 330 (91%) | 58 (75%) | | 129 (88%) | 31 (76%) | |
| THA | 32 (9%) | 19 (25%) | | 18 (12%) | 10 (24%) | |

(*Continued*)

**Table 2.** (Continued)

| Characteristics | Training Cohort | | | Internal Test Cohort | | |
|---|---|---|---|---|---|---|
| | Non-complication N = 362 | Complication N = 77 | p-value | Non-complication N = 147 | Complication N = 41 | p-value |
| **Anesthesia method** | | | 0.931 | | | >0.999 |
| General anesthesia | 55 (15%) | 12 (16%) | | 17 (12%) | 5 (12%) | |
| Combined spinal and epidural anesthesia | 307 (85%) | 65 (84%) | | 130 (88%) | 36 (88%) | |

retained relevant evaluation variables, has highest accuracy for preoperative frailty screening [19], and easy to collect data in clinical. Therefore, we choose mFI-5 as the variable to research.

The mFI-5 is a prognostic tool that incorporates five factors: hypertension, congestive heart failure, chronic obstructive pulmonary disease (COPD), diabetes, and functional dependency prevalence. An mFI-5 score ≥3 indicates frailty [20], providing a holistic assessment of patient physiological status by integrating data from the respiratory, circulatory, endocrine, nervous, and musculoskeletal systems [21]. This non-specific indicator is valuable for early identification of frailty risk in elderly surgical patients and can forecast adverse perioperative outcomes, including postoperative falls, infections, and mortality [22]. The mFI-5 has found broad application across various surgical settings. Lim's research highlighted frailty's impact on prognosis post-coronary artery bypass grafting, influencing recovery of cardiac function and extending hospital stays [23]. Huang's study underscored mFI-5's high sensitivity in predicting severe complications, prolonged hospitalization, and mortality following curative resection of colorectal cancer [24]. Ma's study involving 294 elderly gynecological patients affirmed that the mFI-5 serves as an independent predictor of postoperative complications [25]. Olson's

**Table 3. The coefficients of Lasso regression analysis.**

| Coefficient | variable |
|---|---|
| -2.6912202 | (Intercept) |
| 0.0000000 | Gender_level_ |
| 0.0000000 | Age_level_ |
| 0.0000000 | BMI_level_ |
| 0.0000000 | Classification.of.fracture_level_ |
| 0.4218567 | mFI.5_level_ |
| 0.0000000 | Blood.transfusion_level_ |
| 0.0000000 | HGB_level_ |
| 0.0000000 | Fibrinogen_level_ |
| 1.0785122 | CAR_level_ |
| 0.0000000 | urea.nitrogen_level_ |
| 0.0000000 | Creatinine_level_ |
| 0.0000000 | Triglyceride_level_ |
| 0.0000000 | Color.Doppler.Ultrasound_level_ |
| 0.0000000 | Operation.time_level_ |
| 0.0000000 | ASA_level_ |
| 0.0000000 | Preoperative.fasting.time_level_ |
| 0.0000000 | Injury.surgery.interval_level_ |
| 0.0000000 | Surgical.method_level_ |
| 0.0000000 | Anesthesia.method_level_ |

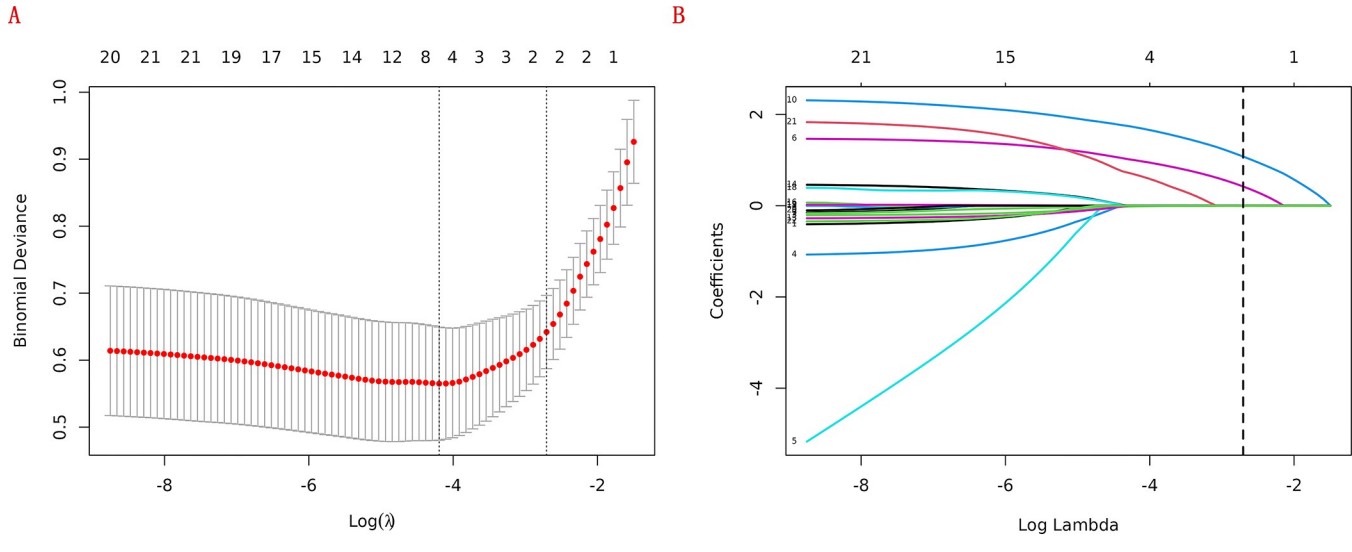

**Fig 1.**

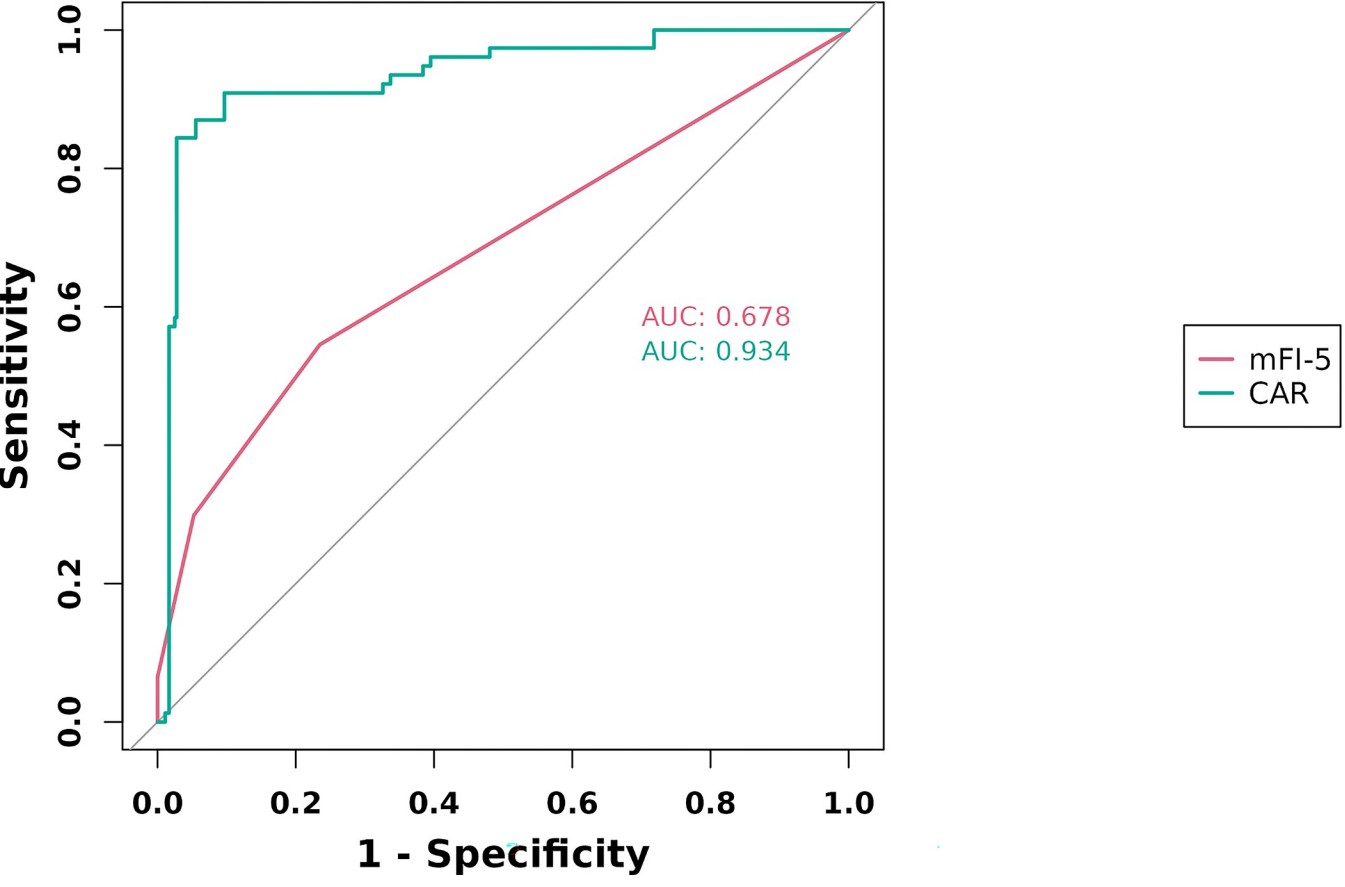

**Fig 2.**

**Table 4. Results of multivariate logistic regression for training cohort.**

| Characteristic | N | Event N | OR | 95% CI[1] | p-value |
|---|---|---|---|---|---|
| mFI-5 | 439 | 77 | 3.49 | 2.27, 5.37 | <0.001 |
| CAR | 439 | 77 | 8.12 | 5.00, 13.21 | <0.001 |

research in orthopedics demonstrated that spinal surgery patients with higher frailty indexes face significantly increased risks of postoperative complications, whereas controlled frailty indicators do not elevate perioperative risks [26]. Dave's findings underscored that frail elderly patients have a notably heightened risk of postoperative deep vein thrombosis and reoperation within 30 days [27]. Kim's extensive study of 7,540 patients following THA validated mFI-5's accuracy in predicting complications such as pulmonary infections, readmission rates, and other post-surgical complications, establishing it as a robust independent predictive tool [28]. Similarly, Shen [29] investigation of 965 elderly hip fracture patients aged 60 to 100, identified a preoperative frailty prevalence of 13.06%, with frailty strongly correlating with the severity of complications. These research findings consistently support the inclusion of mFI-5 in preoperative planning across various surgical specialties.

Some studies suggest that the mFI-5 may have minimal impact on the hospital stay of elderly patients undergoing surgical treatment for hip fractures, possibly due to effective preoperative management [30]. Notably, delirium, another area extensively studied with mFI-5, was not assessed in this study. At our institution, comprehensive preoperative education for elderly patients includes cognitive training, family support, and enhanced sleep management with dexmedetomidine when necessary during surgery. Additionally, psychological counseling, nutritional support, and postoperative functional recovery programs are provided. Consequently, the incidence of postoperative delirium is minimized, and hospital stays are not prolonged as a result. In this study, specific complications such as lower limb deep vein thrombosis were prevalent. Several factors contribute to this high incidence: 1. Patients with comorbidities such as cardiovascular and pulmonary diseases, as well as diabetes, often exhibit increased blood viscosity [31]. 2. The combination of hip fracture and surgical trauma triggers platelet activation, vascular endothelial damage, fibrinolysis activation, and other coagulation-related changes, alongside activation of the complement cascade and innate immune response due to tissue injury. 3. Postoperative fluid deficit. 4. Lower limb edema exacerbates these risks.

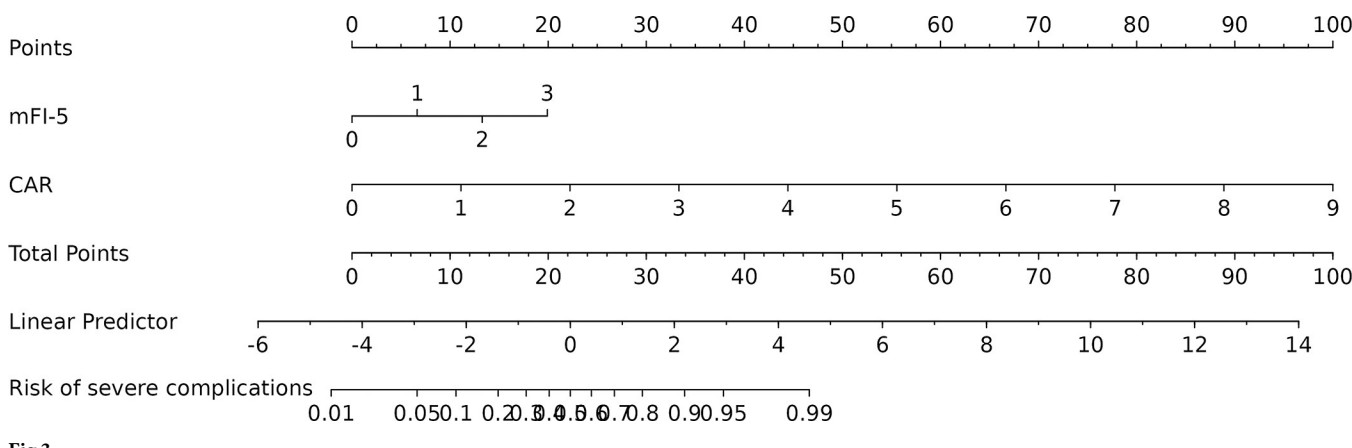

**Fig 3.**

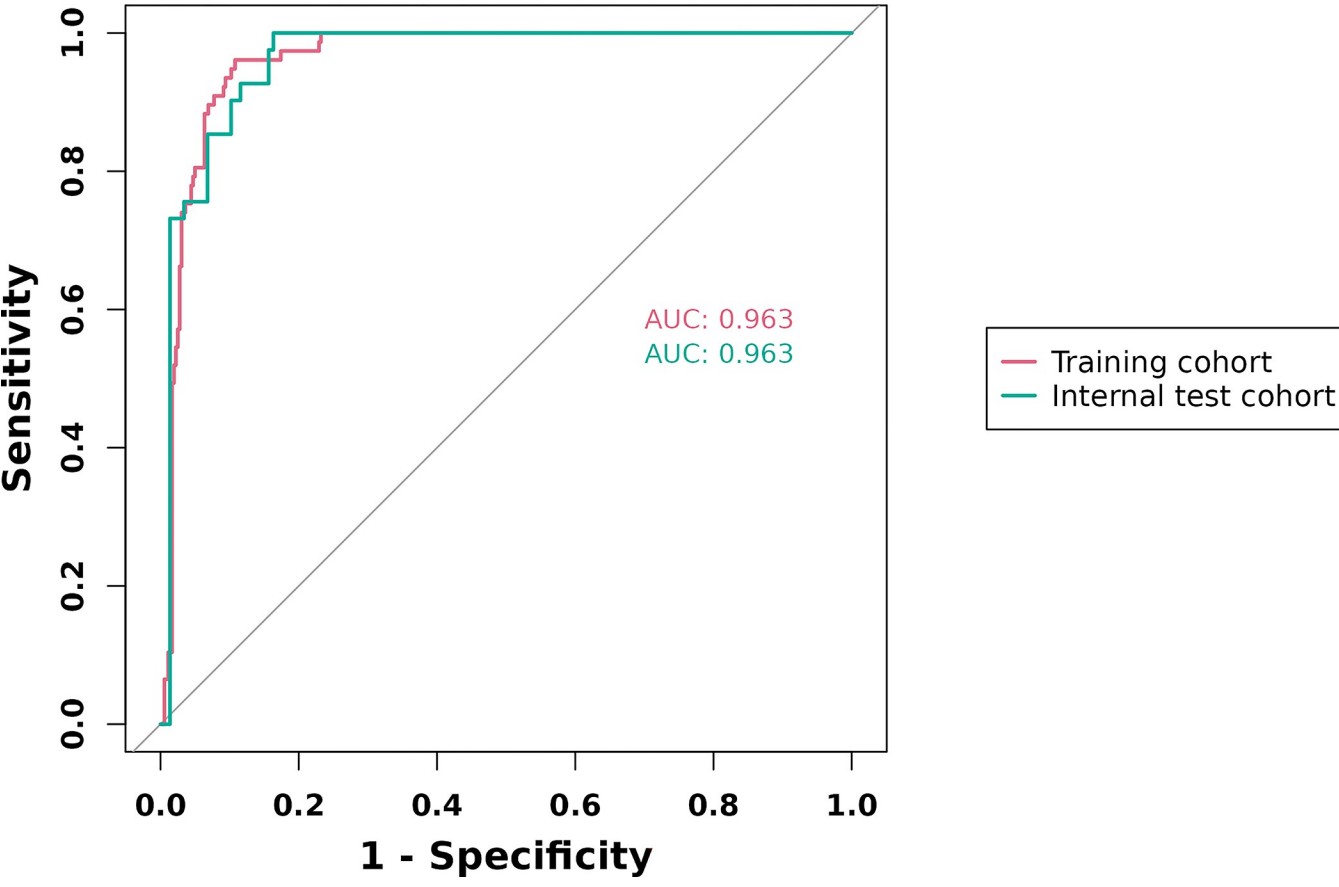

**Fig 4.**

CRP is synthesized by the liver in response to infection, inflammation, and tissue damage, acting as a specific inflammatory indicator [32]. It reflects the patient's post-injury stress state and the effect of postoperative recovery. High CRP levels have been linked to various postoperative complications. Early studies indicated that elevated preoperative CRP levels significantly increased the risk of infection, cardiovascular, and cerebrovascular complications [33]. Research by Song demonstrated a positive correlation between CRP levels and postoperative delirium. Specifically, when postoperative CRP exceeded 100 mg/L, the incidence of complications and mortality rose [34]. A systematic review suggested that while CRP cannot independently diagnose infection, it serves as an indicator necessitating follow-up and analysis of pre- and post-surgery CRP levels to understand underlying causes [35]. Research by Kim concluded that elevated preoperative CRP levels, particularly >1000 mg/L, are associated with increased one-year mortality in elderly patients following hip fracture surgery [36].

Serum albumin levels serve as a critical clinical marker for assessing nutritional status in elderly patients undergoing hip fracture surgery. Variations in albumin levels, influenced by surgical stress, blood loss, immobilization, and poor nutritional intake, significantly impact patient outcomes [37]. In a retrospective analysis involving 278 elderly hip fracture patients, Wang observed that decreased perioperative serum albumin levels and increased C-reactive protein (CRP) levels were associated with adverse postoperative events [38]. Residori's prospective study demonstrated that preoperative serum albumin levels below 30g/L posed a significantly higher risk of postoperative complications, irrespective of the surgical approach—

A

B

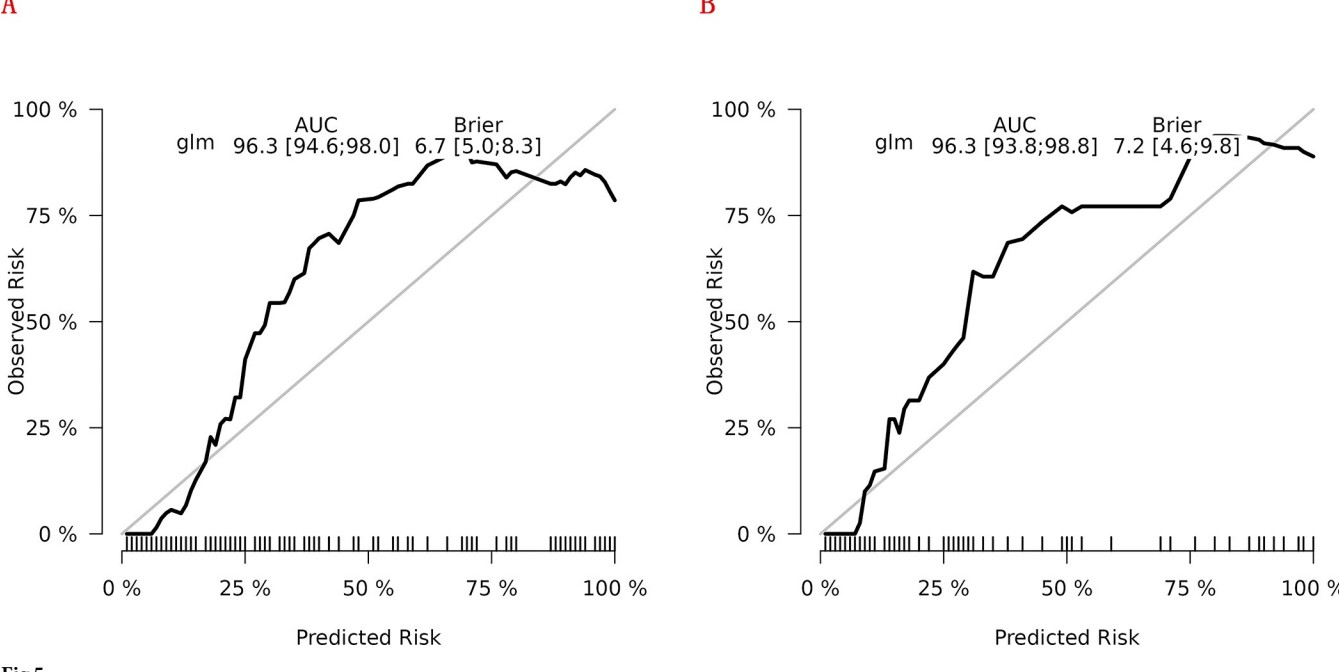

**Fig 5.**

internal fixation or arthroplasty [39]. Subsequently, Wang's risk prediction model, based on 720 elderly hip fracture patients, identified preoperative hypoalbuminemia as an independent risk factor for postoperative pulmonary infections [40]. Kieffer's research linked preoperative

A

B

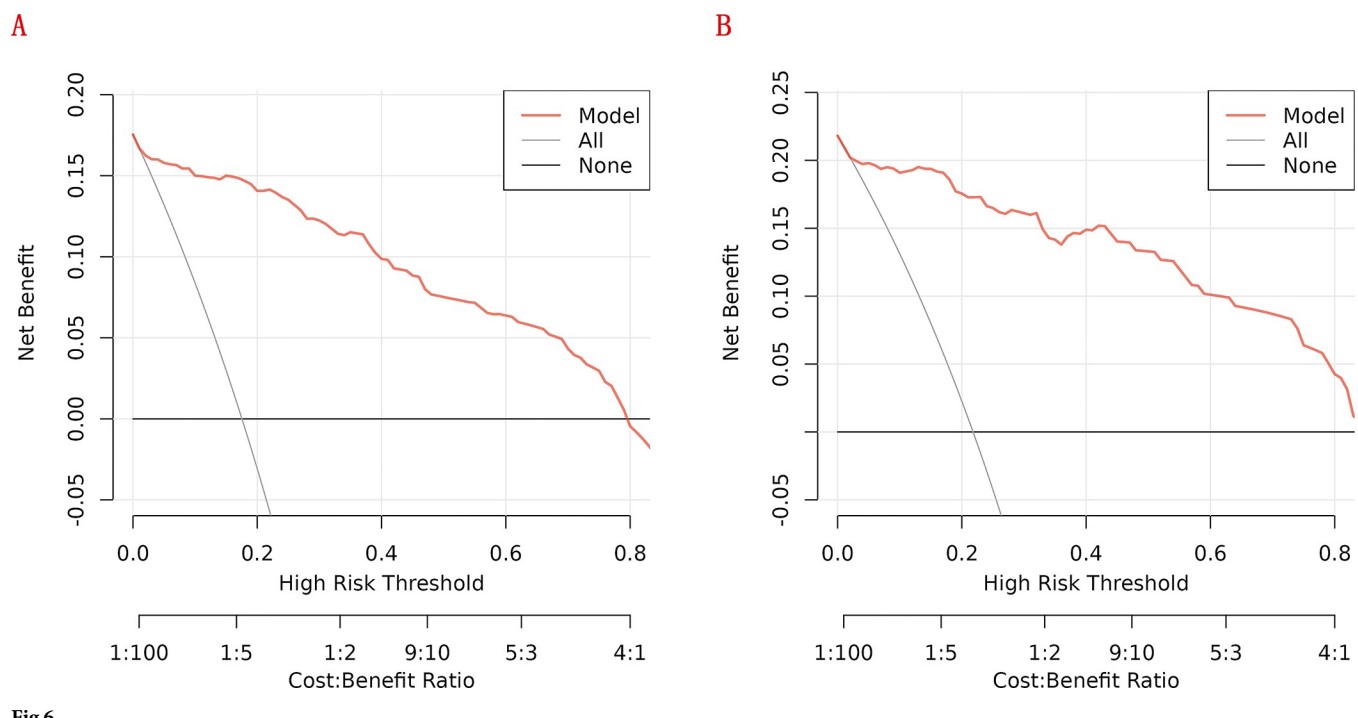

**Fig 6.**

hypoalbuminemia with prolonged hospital stays and increased one-year mortality rates in this patient population [41].

While previous research has often concentrated on single factors, this study integrates the CRP-to-albumin ratio (CAR) to offer a more comprehensive preoperative assessment. CAR is increasingly recognized as a prognostic indicator useful for assessing the prognosis of malignant tumors and predicting perioperative complications [42]. In orthopedics, CAR has been identified as an independent risk factor for postoperative complications and mortality in elderly hip fracture patients, demonstrating high sensitivity and specificity [43], which aligns with the findings presented here.

Limitations of this study include its retrospective nature and relatively small sample size, which may introduce bias and instability in the findings. Future research should prioritize larger-sample, multicenter, randomized controlled trials to validate these results. Additionally, while our predictive model and nomogram incorporate established risk factors, there may be unidentified variables influencing postoperative complications following hip fracture, necessitating further investigation and validation. The applicability of the model developed here is limited to surgical patients, and its generalizability to non-surgical patients requires additional exploration. Furthermore, although various postoperative complications were segmented and analyzed statistically, such as pulmonary infections, venous thromboses, heart failure, renal failure, gastrointestinal symptoms, and urinary tract infections, the model's integration of multiple indicators without detailed analysis of each specific complication may impact prediction accuracy.

## Conclusion

This study confirms that mFI-5 and CAR are predictive of severe postoperative complications in elderly patients undergoing hip fracture surgery. The nomogram model, which integrates these indicators, demonstrates robust predictive capability for severe postoperative complications in this elderly cohort following hip fracture surgery.

## Supporting information

**S1 Dataset. Minimal data set.**
(XLS)

## Acknowledgments

Authors would like to thanks the PLOS ONE's editor and reviewers for their valuable comments and suggestions that helped to enhance the quality of the manuscript.

## Author Contributions

**Conceptualization:** Zhihui Wei, Lian Jiang.

**Data curation:** Zhihui Wei, Lian Jiang.

**Formal analysis:** Zhihui Wei, Lian Jiang.

**Funding acquisition:** Zhihui Wei, Lian Jiang.

**Investigation:** Zhihui Wei, Lian Jiang.

**Methodology:** Zhihui Wei, Lian Jiang, Xiao Chen.

**Project administration:** Zhihui Wei, Lian Jiang, Minghua Zhang.

**Resources:** Zhihui Wei, Lian Jiang, Minghua Zhang.

**Software:** Zhihui Wei, Lian Jiang, Minghua Zhang.

**Supervision:** Zhihui Wei, Lian Jiang, Minghua Zhang, Xiao Chen.

**Validation:** Zhihui Wei, Lian Jiang, Minghua Zhang, Xiao Chen.

**Visualization:** Zhihui Wei, Lian Jiang, Minghua Zhang.

**Writing – original draft:** Zhihui Wei, Lian Jiang, Xiao Chen.

**Writing – review & editing:** Zhihui Wei, Lian Jiang, Xiao Chen.

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
