## [Decision Letter · Decision Letter 0]

1 Aug 2024

PONE-D-24-28136

Development and Validation of a Risk Prediction Model for Severe Postoperative Complications in Elderly Patients with Hip Fracture

PLOS ONE

Dear Dr. Chen,

Thank you for submitting your manuscript to PLOS ONE. After careful consideration, we feel that it has merit but does not fully meet PLOS ONE’s publication criteria as it currently stands. Therefore, we invite you to submit a revised version of the manuscript that addresses the points raised during the review process.

We look forward to receiving your revised manuscript.

Kind regards,

Barry Kweh

Academic Editor

PLOS ONE

Journal Requirements:

Additional Editor Comments:

An interesting study based upon novel data which requires a broader discussion of other frailty index stratification tools and a stronger argument as to why the authors' instruments is superior to others.

Reviewers' comments:

Reviewer's Responses to Questions

**Comments to the Author**

1. Is the manuscript technically sound, and do the data support the conclusions?

Reviewer #1: Yes

Reviewer #2: No

2. Has the statistical analysis been performed appropriately and rigorously? 

Reviewer #1: No

Reviewer #2: No

3. Have the authors made all data underlying the findings in their manuscript fully available?

Reviewer #1: Yes

Reviewer #2: No

4. Is the manuscript presented in an intelligible fashion and written in standard English?

Reviewer #1: Yes

Reviewer #2: No

5. Review Comments to the Author

Reviewer #1: Reviewer Comments:

This study summarizes and statistically analyzes 390 hip fracture surgery patients to identify potential risk factors for postoperative complications, concluding that mFI-5 and CAR are predictors of severe complications in hip fracture patients. This study has clinical significance and is a valuable paper. The following modifications are recommended:

1.This study included a total of 390 cases and was divided into a training group and a validation group. For epidemiological research, the sample size is slightly small.

2.This study is a single-center study. Can the patient population at your institution represent the entire population of patients with this disease?

3.For hip fracture patients, it is recommended to perform detailed classification rather than just dividing them into femoral neck fractures and intertrochanteric fractures. I believe that the severity of the fracture also has a significant impact on postoperative complications. For example, type IV comminuted fractures should also be analyzed as predictive factors.

4.This study is a retrospective study that selected inpatient case data. How was the grouping of cases determined? If not randomized, the statistical results in the training group may be biased.

5.What is the observation period for severe complications in this study? Only retrospective analysis of in-hospital data—are these data sufficient? What was your average hospital stay?

6.Additionally, for severe complications that occurred during hospitalization, your results show that they account for 1/4 of the total cases. This proportion seems a bit high.

Reviewer #2: This study aimed to investigate risk factors associated with severe postoperative complications following hip fracture surgery in elderly patients and to develop a nomogram-based risk prediction model for these complications. They found both the mFI-5 and CAR are predictive factors for severe postoperative complications in elderly patients undergoing hip fracture surgery.This study is not novel enough to be accepted.

6. PLOS authors have the option to publish the peer review history of their article (what does this mean?). If published, this will include your full peer review and any attached files.

Reviewer #1: No

Reviewer #2: No

---

## [Author Response · Author response to Decision Letter 0]

16 Aug 2024

Dear Reviewer,

I appreciate your valuable input. I will make the necessary revisions as per your guidance and eagerly await your next communication.

Reviewer #1: Reviewer Comments:

This study summarizes and statistically analyzes 390 hip fracture surgery patients to identify potential risk factors for postoperative complications, concluding that mFI-5 and CAR are predictors of severe complications in hip fracture patients. This study has clinical significance and is a valuable paper. The following modifications are recommended:

1.This study included a total of 390 cases and was divided into a training group and a validation group. For epidemiological research, the sample size is slightly small.

Answer: Our research team, acknowledging the initial limited sample size, expanded our data collection period to January 2015 through April 2024. The updated dataset now includes 627 cases, with 439 in the training set and 188 in the validation set, markedly increasing the sample size and bolstering the study's reliability. See Tables 1-2 for details.

2.This study is a single-center study. Can the patient population at your institution represent the entire population of patients with this disease?

Answer: The findings of this study may not be consistent with data from all medical institutions due to varying perioperative management practices, which can lead to differences in complication rates. Our focus was on severe postoperative complications following hip fracture surgery, such as septic shock, deep surgical site infections, and lower extremity deep vein thrombosis. For example, the incidence of lower extremity deep vein thrombosis, reported in a substantial body of literature as ranging from 8% to 15%, was approximately 12% in our study, aligning with existing data. The incidence of other complications was also found to be similar to that reported in previous research. Consequently, this study can be considered representative of data from the majority of medical institutions. 

3.For hip fracture patients, it is recommended to perform detailed classification rather than just dividing them into femoral neck fractures and intertrochanteric fractures. I believe that the severity of the fracture also has a significant impact on postoperative complications. For example, type IV comminuted fractures should also be analyzed as predictive factors.

Answer: The influence of fracture severity on the occurrence of postoperative complications is indeed a valuable observation. We intend to conduct a dedicated investigation into this aspect in future research. Thank you for your insightful comment. 

4.This study is a retrospective study that selected inpatient case data. How was the grouping of cases determined? If not randomized, the statistical results in the training group may be biased.

Answer: Following the compilation of all patient records, we employed R 4.2.2 software to perform random grouping at a 7:3 ratio, distinguishing between a training group and a validation group. A comparison of variables was conducted prior to advancing to the next phase of statistical analysis. For comprehensive details, consult the statistical section of the original paper's methodology. This procedure guaranteed compliance with the principle of data randomization, which is crucial, as failure to randomize could introduce bias into the statistical results of both groups, thereby significantly impacting the findings.

5.What is the observation period for severe complications in this study? Only retrospective analysis of in-hospital data—are these data sufficient? What was your average hospital stay?

Answer: Our assessment encompasses a hospital stay of 8.5 days, which constitutes our observation period. The period of 3 to 5 days post-surgery is identified as the time of highest risk for the development of severe complications. The statistical data we have gathered is focused primarily on patients admitted during this critical hospitalization phase.

6.Additionally, for severe complications that occurred during hospitalization, your results show that they account for 1/4 of the total cases. This proportion seems a bit high.

Answer: The elevated incidence of severe postoperative complications in this study may be attributed to the following factors:

(1)An initially small sample size led us to expand our data collection timeframe to January 2015 through April 2024. The revised dataset encompasses 627 cases, of which 118 developed severe postoperative complications, representing about 18.82% of the total.

(2)The majority of elderly patients with hip fractures admitted to our hospital are transferred from lower-tier hospitals for advanced treatment. These patients frequently present with multiple comorbidities and more severe conditions, potentially contributing to the higher relative rate of postoperative complications.

Reviewer #2: This study aimed to investigate risk factors associated with severe postoperative complications following hip fracture surgery in elderly patients and to develop a nomogram-based risk prediction model for these complications. They found both the mFI-5 and CAR are predictive factors for severe postoperative complications in elderly patients undergoing hip fracture surgery.This study is not novel enough to be accepted.

Answer:Thank you for your comments, I will revise this study.

---

## [Editor Report · Decision Letter 1]

22 Aug 2024

PONE-D-24-28136R1Development and validation of a risk prediction model for severe postoperative complications in elderly patients with hip fracturePLOS ONE

Dear Dr. Chen,

Thank you for submitting your manuscript to PLOS ONE. After careful consideration, we feel that it has merit but does not fully meet PLOS ONE’s publication criteria as it currently stands. Therefore, we invite you to submit a revised version of the manuscript that addresses the points raised during the review process.

We look forward to receiving your revised manuscript.

Kind regards,

Barry Kweh

Academic Editor

PLOS ONE

Additional Editor Comments:

This is a retrospective cohort with its inherent weaknesses but provides original data on the utility of risk prediction in a vulnerable group of patients. The authors have revised their paper to strengthen the discussion.

---

## [Author Response · Author response to Decision Letter 1]

26 Aug 2024

Dear Reviewer,

I appreciate your valuable input. I will make the necessary revisions as per your guidance and eagerly await your next communication.

Reviewer #1: Reviewer Comments:

This study summarizes and statistically analyzes 390 hip fracture surgery patients to identify potential risk factors for postoperative complications, concluding that mFI-5 and CAR are predictors of severe complications in hip fracture patients. This study has clinical significance and is a valuable paper. The following modifications are recommended:

1.This study included a total of 390 cases and was divided into a training group and a validation group. For epidemiological research, the sample size is slightly small.

Answer: Our research team, acknowledging the initial limited sample size, expanded our data collection period to January 2015 through April 2024. The updated dataset now includes 627 cases, with 439 in the training set and 188 in the validation set, markedly increasing the sample size and bolstering the study's reliability. See Tables 1-2 for details.

2.This study is a single-center study. Can the patient population at your institution represent the entire population of patients with this disease?

Answer: The findings of this study may not be consistent with data from all medical institutions due to varying perioperative management practices, which can lead to differences in complication rates. Our focus was on severe postoperative complications following hip fracture surgery, such as septic shock, deep surgical site infections, and lower extremity deep vein thrombosis. For example, the incidence of lower extremity deep vein thrombosis, reported in a substantial body of literature as ranging from 8% to 15%, was approximately 12% in our study, aligning with existing data. The incidence of other complications was also found to be similar to that reported in previous research. Consequently, this study can be considered representative of data from the majority of medical institutions. 

3.For hip fracture patients, it is recommended to perform detailed classification rather than just dividing them into femoral neck fractures and intertrochanteric fractures. I believe that the severity of the fracture also has a significant impact on postoperative complications. For example, type IV comminuted fractures should also be analyzed as predictive factors.

Answer: The influence of fracture severity on the occurrence of postoperative complications is indeed a valuable observation. We intend to conduct a dedicated investigation into this aspect in future research. Thank you for your insightful comment. 

4.This study is a retrospective study that selected inpatient case data. How was the grouping of cases determined? If not randomized, the statistical results in the training group may be biased.

Answer: Following the compilation of all patient records, we employed R 4.2.2 software to perform random grouping at a 7:3 ratio, distinguishing between a training group and a validation group. A comparison of variables was conducted prior to advancing to the next phase of statistical analysis. For comprehensive details, consult the statistical section of the original paper's methodology. This procedure guaranteed compliance with the principle of data randomization, which is crucial, as failure to randomize could introduce bias into the statistical results of both groups, thereby significantly impacting the findings.

5.What is the observation period for severe complications in this study? Only retrospective analysis of in-hospital data—are these data sufficient? What was your average hospital stay?

Answer: Our assessment encompasses a hospital stay of 8.5 days, which constitutes our observation period. The period of 3 to 5 days post-surgery is identified as the time of highest risk for the development of severe complications. The statistical data we have gathered is focused primarily on patients admitted during this critical hospitalization phase.

6.Additionally, for severe complications that occurred during hospitalization, your results show that they account for 1/4 of the total cases. This proportion seems a bit high.

Answer: The elevated incidence of severe postoperative complications in this study may be attributed to the following factors:

(1)An initially small sample size led us to expand our data collection timeframe to January 2015 through April 2024. The revised dataset encompasses 627 cases, of which 118 developed severe postoperative complications, representing about 18.82% of the total.

(2)The majority of elderly patients with hip fractures admitted to our hospital are transferred from lower-tier hospitals for advanced treatment. These patients frequently present with multiple comorbidities and more severe conditions, potentially contributing to the higher relative rate of postoperative complications.

Reviewer #2: This study aimed to investigate risk factors associated with severe postoperative complications following hip fracture surgery in elderly patients and to develop a nomogram-based risk prediction model for these complications. They found both the mFI-5 and CAR are predictive factors for severe postoperative complications in elderly patients undergoing hip fracture surgery.This study is not novel enough to be accepted.

Answer:Thank you for your comments, I will revise this study.

Additional Editor Comments:

This is a retrospective cohort with its inherent weaknesses but provides original data on the utility of risk prediction in a vulnerable group of patients. The authors have revised their paper to strengthen the discussion.

Answer:Thank you for your comments, I will strengthen the discussion.

---

## [Editor Report · Decision Letter 2]

2 Sep 2024

Development and validation of a risk prediction model for severe postoperative complications in elderly patients with hip fracture

PONE-D-24-28136R2

Dear Dr. Chen,

We’re pleased to inform you that your manuscript has been judged scientifically suitable for publication and will be formally accepted for publication once it meets all outstanding technical requirements.

Kind regards,

Barry Kweh

Academic Editor

PLOS ONE

Additional Editor Comments (optional):

The authors have clarified their dataset and statistical analysis as well as now including baseline surgical and medical complication rates in their manuscript to strengthen the discussion.
---

## [Editor Report · Acceptance letter]

3 Sep 2024

PONE-D-24-28136R2 

PLOS ONE

Dear Dr. Chen, 

I'm pleased to inform you that your manuscript has been deemed suitable for publication in PLOS ONE. Congratulations! Your manuscript is now being handed over to our production team.

Kind regards, 

on behalf of

Dr. Barry Kweh 

Academic Editor

PLOS ONE